# Lightweight Crypto-Ransomware Detection in Android Based on Reactive Honeyfile Monitoring

**DOI:** 10.3390/s24092679

**Published:** 2024-04-23

**Authors:** José A. Gómez-Hernández, Pedro García-Teodoro

**Affiliations:** Network Engineering & Security Group (NESG), School of Computer Science and Telecommunications Engineering (ETSIIT), C/Periodista Daniel Saucedo Aranda s/n, 18071 Granada, Spain; pgteodor@ugr.es

**Keywords:** crypto-ransomware, early detection, deception, reactive monitoring, honeyfile, Android

## Abstract

Given the high relevance and impact of ransomware in companies, organizations, and individuals around the world, coupled with the widespread adoption of mobile and IoT-related devices for both personal and professional use, the development of effective and efficient ransomware mitigation schemes is a necessity nowadays. Although a number of proposals are available in the literature in this line, most of them rely on machine-learning schemes that usually involve high computational cost and resource consumption. Since current personal devices are small and limited in capacities and resources, the mentioned schemes are generally not feasible and usable in practical environments. Based on a honeyfile detection solution previously introduced by the authors for Linux and Window OSs, this paper presents a ransomware detection tool for Android platforms where the use of trap files is combined with a reactive monitoring scheme, with three main characteristics: (i) the trap files are properly deployed around the target file system, (ii) the *FileObserver* service is used to early alert events that access the traps following certain suspicious sequences, and (iii) the experimental results show high performance of the solution in terms of detection accuracy and efficiency.

## 1. Introduction

Security incidents impact ICT-related systems and services and, therefore, society in general [1,2,3,4]. In particular, any kind of information and communication infrastructure, such as those relying on 5/6G and IoT technologies, is continuously exposed to such threats [5,6,7,8,9,10,11].

In that global context, and despite its years of existence, ransomware continues to be a principal specific security threat [12,13,14]. As indicated in a number of works like [15,16,17], ransomware is mainly focused on stealing user data (mainly by ciphering it) and affects both individuals and organizations of any type (e.g., hospitals, banks, industry, energy, etc.) all over the world. Moreover, the effects are diverse and range from economical and reputational losses in companies and organizations to health-related and mental problems in individuals [18,19,20,21].

Provided the relevance and impact of ransomware and the current generalized adoption of mobile devices among users (e.g., smartphones, tablets, and IoT-related devices), researchers have developed a number of ransomware mitigation proposals for these types of environments, in particular for the Android platform since it is the most used OS today [22,23,24,25].

Despite the good accuracy rates reached by current ransomware detection schemes, they usually rely on machine-learning techniques [26,27,28,29], which involves high resource consumption in terms of computation complexity and memory/CPU/battery usage. In fact, works like [17,30,31] point out the need for early detection and low resource consumption, especially for limited devices like the IoT-related ones as, otherwise, the solutions may not be feasible or usable.

Based on previous authors’ work [32,33], this paper presents a new ransomware detection tool for Android platforms with three main contributions:It is a deception-based approach, where a set of decoy files are deployed around the file system to capture the attacker’s attention. The proposal takes advantage of the fact that any ransomware sample must access and manipulate files (either legitimate or traps) to kidnap them.The detection process itself relies on a reactive file system monitoring solution that fires alarms when trap files are accessed or changed following a certain suspicious sequence.Since the (reactive) solution does not run as long as the traps are not accessed, the detection process itself does not consume significant resources in terms of memory, computation, CPU usage, etc. In other words, a high detection accuracy as well as a lightweight, low consumption detection solution is here provided.

The rest of the paper is organized as follows. Section 2 presents and discusses several relevant and recent contributions on the topic of Android ransomware detection. Section 3 introduces the detection paradigm relying on the use of trap files to thwart malicious activities by fooling the attacker. In particular, our specific previous work for Linux and Windows platforms is here presented. Then, in Section 4 we discuss the limitations of Android to adapt the previous solution to this particular OS, so that the use of decoy files is complemented with a reactive file system monitoring detection scheme based on the class FileObserver() (using inotify) to fire an alert after files are accessed or changed. Section 5 presents a proof of concept experimentation to evaluate the validity of our proposal, the results showing a good performance of it in terms of high accuracy and low resource consumption as expected. Finally, Section 6 highlights the main contributions and future work.

## 2. Background of Ransomware Detection for Android Platforms

Data kidnapping may be dramatic for both organizations and individuals, as the affected information can be highly sensitive from several perspectives (e.g., health, sex, religion, intimacy, etc.). This situation is especially relevant in new mobility environments where user devices (smartphones, biosensors, etc.) are widely utilized for both work and personal operations. As discussed in the previous section, this is the reason why considerable efforts are made by the community to combat crypto-ransomware. In this section, we comment on some of the most significant and current proposals in the literature in this field.

Razgallah et al. surveyed in [34] the principal methods and mechanisms for general malware detection in Android applications, which are also applicable to ransomware. On the one hand, detection can be static or dynamic. In the first case, a target software/program is analyzed prior to its execution in order to discern its malicious or benign behavior. Instead, dynamic detection need to execute the software/program to analyze its behavior and impact on the system. On the other hand, static schemes can be generally classified into three categories: (i) related with application code (whether at source level or bytecode level); (ii) related on permissions requested and different API calls embedded in the app code; (iii) other methods combining multiple factors. For their part, dynamic detection techniques can be organized into four categories: (i) relying on system calls; (ii) methods based on other information at the system level like CPU consumption or network use; (iii) schemes that depend on user space level information like API calls; (iv) techniques that observe the dynamic behavior of the app user other than system calls or information at the level of system or user.

In this general context, we present in the rest of the section specific detection proposals developed in the literature, as described by the authors. Scalas et al. proposed in [35] learning-based detection methods from information of the API used in ransomware attacks when performing their actions. The authors tested three forms employing this information: packages, classes, and methods. The comparison of performance obtained with other approaches showed that this method can detect not only ransomware but also general malware apps with high accuracy, as good as other methods, all in less than 0.2 s. Additionally, this systems can detect novel samples and is resilient against static obfuscation attempts.

Alzahrani et al. introduced in [36] RanDetector, a slight and automated detector for Android ransomware apps according to their behavior. The detection system investigated information like permissions, intents, and APIs before classify the app by integrating different supervised machine-learning models. The system obtained a detection rate superior to 97.62%, with almost zero false positive at a time cost of ∼45 s. Again making use of permissions and APIs, the authors in [37] proposed a hybrid detection system with high accuracy. In the static analysis, more than 70 widely accepted antivirus engines were considered. In the dynamic analysis, a comparative study was conducted to find the correct tool for integrated it into the ransomware detection. The experimentation showed that static analysis have approximately a 40–55% detection accuracy regarding 100% of dynamic analyses. Also as a hybrid approach, Arora and Kumar combined static with dynamic detection to introduce a ransomware detection toolkit in [38]. The static features (permissions and APIs) were passed through an artificial neural network, while the dynamic ones (network traffic) were passed through an LGBM classifier to detect ransomware on the network.

The authors in [39] used the algorithms Random Forest, J48, and Naïve Bayes for dynamic detection. More innovative, Chen et al. introduced RansomProber in [40], where user interface widgets and the coordinates of user keystrokes were analyzed to derive potential operations related to the encryption of files. The results showed great accuracy and a good runtime performance (∼5 s time cost and 19 MB memory usage).

Faghihi and Zulkernine presented in [41] RansomCare, a proposal based on the structure of user data and their entropy for ransomware detection and mitigation. RansomCare was able not only to detect but also to neutralize crypto-ransomware in real-time in just 1.4 s with dynamic and static analysis. The detection proposal was capable of recovering lost files while maintaining privacy based on monitoring changes in user data.

Manzil and Naik proposed in [42] a technique of feature selection based on hamming distance for the static analysis of ransomware detection. The approach involved four steps: feature extraction (like permissions and intents), generation of a binary vector of features, their selection, and finally its classification. The detection accuracy achieved was 99% using Random Forest and Decision Tree classifiers, while the involved complexity was $O(n^{2})$.

Sharma et al. proposed in [43] a detection method where new ransomware features were used, the feature dimensionality was reduced, an ensemble learning model for detection was employed, and a comparative analysis to identify the computational time of detection was conducted. The results indicated that detection accuracy was 99.67% with the Random Forest ensemble model. The feature dimensionality reduction through the principal component analysis method ensured that the Logistic Regression model had a lower execution time on GPU than on CPU.

The same authors performed in [44] a deep analysis of ransomware to extract static features through reverse engineering and forensic analysis from the apk file and source code, respectively. Additionally, a RansomDroid framework based on the Gaussian Mixture Model was proposed. By using feature selection combined with dimensionality reduction, the experimental accuracy on Android ransomware detection was 98.08%, taking 44 ms.

In [45], Almomani et al. proposed an efficient detection approach based on machine learning. The approach focused on version 11 of Android and API Level 30, to obtain the most recent set of features, like permissions and API calls, that ransomware could potentially make use of. Afterwards, different predictive models for ransomware were generated using different machine-learning techniques like Random Forest, Decision Tree, Sequential Minimal Optimization, and Naive Bayes. Some of them presented an accuracy of 98.3%, even after reducing the feature set by approximately 26%. Likewise, the authors in [46] introduced a ransomware detection methodology that rested on an evolutionary machine-learning technique, where the tuning of hyper-parameters for the classification algorithm was performed with a binary Particle Swarm Optimization (PSO) algorithm. Classification was made with the Support Vector Machine (SVM) algorithm combined with the Synthetic Minority Oversampling Technique (SMOTE). The performance of the SMOTE–tBPSO–SVM method was better than traditional algorithms.

The authors in [47] described an Android ransomware detection method that used PSO to select (84) traffic features. The data traffic was classified with the Decision Tree and Random Forest classifiers. The latter achieved the greats performance in detection, whereas the former was the best for detecting ransomware types.

Ahmed and Al-Dabbagh analyzed in [48] six machine-learning methods to defend mobile devices from malware by monitoring network traffic: Random Forest, k-Nearest Neighbors, Multi-Layer Perceptron, Decision Tree, Logistic Regression, and eXtreme Gradient Boosting. A similar work is one by Bagui and Woods [28], where the algorithms considered were Decision Tree, Naïve Bayes, and OneR. Again, Jose et al. analyzed in [49] various machine-learning algorithms combining RansomDroid and concept drift in the classification of raw data considering host, network, behavior, and files.

The authors in [50] proposed ARdetector, an architecture for Android ransomware detection that allows the analysis of some related ransomware features, like behavioral characteristics, to select the most representative ones. In addition, a deep neural network using focal loss was designed. Again focusing on machine-learning techniques, the authors in [51] combined static analysis and machine-learning techniques for predicting ransomware applications. For classification, the Decision Tree, Extra Tree classifier, and Light Gradient Boosting Machine methods in concurrence with the Random Forest Tree scheme were employed.

Ngirande et al. proposed in [52] a hybrid analysis employing the Support Vector Machine (SVM) algorithm for Android ransomware detection. Static features as well as dynamic features were used. This model achieved a suitable performance: with static features the accuracy was 81% and the precision was 90%; with dynamic ones, the accuracy was 100%. Ahmed et al. also made use of different techniques in [53] to build efficient, precise, and robust models, including Decision Tree, Support Vector Machine, k-Nearest Neighbor, Ensemble of Decision Tree, Feedforward Neural Network, and tabular attention network for binary classification.

As shown, most of the current Android ransomware detection solutions rely on machine-learning methods. However, they usually focus on detection accuracy and avoid providing computation cost and resource consumption figures, which is a principal concern for user device, as pointed out in works like [17,30,31].

In what follows, we introduce a new crypto-ransomware detection solution for Android platforms based on the use of the deployment of honeyfiles and the reactive monitoring of them over time based on the class FileObserver(), which uses inotify() to alarm events that access the file system. In addition to the detection efficiency and efficacy demonstrated by our approach, its reactive nature involves low resource consumption (almost zero) compared to the active detection schemes usually considered in current detection solutions (either static, dynamic, or hybrid [54]).

## 3. Ransomware Detection Based on Honeyfiles

Deception technology is a category of cybersecurity solutions aimed at detecting threats in a proactive way by the deployment of realistic decoys (e.g., domains, databases, directories, servers, apps, etc.) in an environment alongside real assets to act as lures. The importance of deception techniques in cybersecurity is clear [55,56,57], since they usually present some main benefits:Attackers waste their time exploring worthless planted assets while you bait them into a trap.The moment an attacker interacts with a decoy, the technology begins gathering information that will be used to generate high-fidelity alerts that reduce dwell time and accelerate incident response. That is, an early detection process with low false positive rates is achieved.This technology generates threat intelligence, stops lateral movement, and orchestrates threat response and containment, all without human supervision.

Although there may be some nuances among them, terms like ‘decoy’, ‘trap’, and ‘honey’ are interchangeably used to indicate resources attractive to hackers whose goal is to generate alerts when they are accessed.

Like honeypots and honeynets (See links like https://www.projecthoneypot.org (accessed on 1 April 2024), and https://www.honeynet.org (accessed on 1 April 2024)), honeyfiles draw the attention of cybercriminals to distance them from their real targets. Honeyfiles are easy to set up and maintain, but there is no guarantee that the attack will reach them. The authors in [58] propose the creation of a large dummy file that will be monitored. The encryption process of this file by ransomware will take time, which allows the detection and protection of the rest of the files (changing the attributes of the remaining files and a list of infected files and another of non-infected ones). Notification filters are used to observe changes in some metadata of the files in the monitored folder (name, last access date, last write date, security, and size). The system does not prevent some files from being affected, since the trap file is generated when changes in other files are detected.

Moussaileb et al. use decoy files distributed throughout the file system (especially in folders not in common use) that serve to count the number of times a thread of a program passes through it [59]. By normalizing this counter by the total number of decoy folders considered, a ransomware alert is launched if a fixed threshold is exceeded. This idea was already presented in [60], where name changes and decoy files were simply monitored to detect the presence of ransomware.

The UNVEIL system [61] generates a virtual environment with the aim of attracting attackers and limiting damage before being detected through honeyfiles. Another work in this line is [62], which investigates the creation and monitoring of the activities of a ransomware using the technique of folders acting as honeypots. The work studies two methods for its implementation: the File Screening Service of Microsoft’s File Server Resource Manager (FSRM), and the EventSentry solution to manipulate Windows security logs.

RWGuard [63] uses decoy files that should not be written for detection. They are monitored along with tracking the behavior of processes regarding their I/O requests (IRP) and file changes (creation, erase, and write operations) in search of malicious behaviors. RansomWall [64] is a multi-layer defense that incorporates trap files as a form of protection against crypto-ransomware. When a malicious process is suspected in the trap layer, the modified files are copied until determining whether they are malign or benign in other layers.

More recently, SentryFS [65] is a specialized file system that strategically distributes trap files. These traps are generated using Natural Language Processing (NLP), and both their content and metadata are constantly updated to appear more attractive to more targeted ransomware samples. With this purpose, SentryFS connects to an anti-ransomware web service to download the latest information on new ransomware strategies. Additionally, files are cloned to avoid directly writing to them in case ransomware goes unnoticed. This would encrypt the clones instead of the actual files.

One more approach in the field is the work by Wang et al. [66], where the authors designed and implemented KRProtector as a solution to detect ransomware and protect files based on decoys for IoT devices without ROOT.

### 3.1. R-Locker: A Particular Honeyfile-Based Ransomware Detection Solution for Linux and Windows

Before addressing our specific detection proposal based on trap files for Android in Section 4, we will now describe the architecture and methodology of our previous approach for detecting crypto-ransomware in Linux and Windows systems: *R-Locker*.

#### 3.1.1. R-Locker Architecture

We assume that a crypto-ransomware sample eventually scans the file system of the infected machine, either randomly or selectively, looking for files to access and encrypt their contents, as described in [67]. Based on this general characteristic, we propose an anti-ransomware solution to create a honeyfile that serves as a trap to capture samples of this typology of malware. This proposal has two main beneficial features, **F** = {F1, F2}, that distinguish the proposal from others:F1.The ransomware sample will be blocked when it accesses the honeyfile without affecting the rest of the file system.F2.In addition to blocking the sample, the malicious access is notified and a countermeasure automatically deployed to address the threat.

The methodology corresponds to the functional architecture shown in Figure 1. The operational procedure is conceptual and, therefore, independent of the specific platform where it is implemented (Linux, Windows, Android, etc.). In addition to the aforementioned anti-ransomware properties, some requirements, **R**, should be satisfied by the solution in order to be valid for real environments:R1.*Effectiveness*: The harmful actions of the ransomware on the system must be null or minimized.R2.*Low consumption*: To be scalable, less resources in terms of CPU, memory, and storage must be consumed.R3.*Clarity*: To be usable by end users, no special privileges for the installation or execution are required.R4.*Transparency*: The rest of the applications and services on the system should not be affected.R5.*Simplicity*: No complex operations must be required to thwart the threat.

Developing an anti-ransomware solution that satisfies the core benefits, **F**, and requirements, **R**, seems like a non-trivial task. In the rest of the section we will briefly describe our proposed implementation for Linux/Unix and Windows systems.

#### 3.1.2. R-Locker for Linux/Unix and Windows

A simple and elegant solution to achieve our objectives while satisfying all established requirements is to use FIFOs (also known as named pipe) [32], which provides a unidirectional interprocess communication channel. A FIFO is created as a permanent named object in the file system and has two interesting and useful properties for our purpose due to its dual nature [68]:As an object in the file system, it is manipulated like a conventional file accessible through the File API (open, close, read, write, etc.). This makes a FIFO visible to ransomware.A FIFO is also a finite-sized communication channel between two processes, the synchronization between them being automatically managed by the operating system, which simplifies the proposed solution.

From the above, our anti-ransomware solution firstly developed for Linux systems operates as indicated in Figure 1, where the trap files are implemented as FIFOs. This way, the detection procedure is as follows (see Figure 3 in [32]):1.First, R-Locker creates a FIFO (mkfifo() system call) that will be the central honeyfile or trap file.2.Secondly, the process will open the channel in write-only mode (O_WRONLY) and write to it the necessary bytes to fill it and block the writing process that will act as a monitor. At this point, the trap is ready and waiting for prey.3.From here on, when an external process (a supposed ransomware) starts reading the trap, it will finally be blocked by the operating system.4.Simultaneously, the writer process, which was stopped, is automatically woken up by the kernel and continues its execution to launch the countermeasures as follows: (i) the identifier of the application that accesses the honeyfile is determined; (ii) the user is notified to kill, if necessary, the corresponding process.

Transferring the previous solution to Windows environments is relatively direct [33] since both operating systems have similar abstractions. However, there are two fundamental differences in FIFO management on Windows compared to Unix systems: (a) files and FIFOs belong to different namespaces in the case of Windows, (b) only one process is allowed to simultaneously read from the FIFO on Windows.

To solve the first difference, we need to connect files and FIFO spaces. Windows symbolic links can be used for that, which allow creating an object in the file system that points to an object in the device or FIFO space. Regarding the second difference, we can create several instances of a FIFO so that if several processes access the FIFO, each of them connects to one of said instances. The instances can be created dynamically as needed, which has the advantage of allowing one to handle ransomware families that use multithreaded processes to optimize file encryption.

As shown in the experimentation carried out in [32,33], a high efficiency and low resource consumption must be remarked for our FIFO-based ransomware detection approach, both for Linux and for Windows. Two additional relevant aspects have contributed to the good performance shown by R-Locker:Dynamic management of honeyfiles. Beyond the general behavior of honeyfiles in R-Locker, an important aspect to consider is their location, that is, how to select and manage the folders in which to place the traps. From the results of the work [67] we conclude that there is not a single order for the selection of folders: some samples make an in-depth selection first, then alphabetically; others make a random selection. Such very different behaviors lead to the conclusion that it is advisable to deploy the traps in all folders in order to achieve complete protection. To do this, instead of replicating the traps, which would force us to have to replicate the monitoring process, we can create links to the central FIFO in each and every folder in the file system.Regarding the selection of files within a folder, ransomware samples make selections according to different criteria. In some cases (e.g., NTFS), entries are returned in alphabetical order. In other cases, files are first prioritized by extension and then selected alphabetically. To address this situation we can create multiple links with names “!..!” and extensions like ‘.doc’, ‘.pdf’, ‘.jpg’, etc. To enhance transparency from the user’s perspective, honeyfiles can also be hidden so that they are invisible to normal user operations.Integrated detection and (semi-)automatic response. As previously described, a countermeasure is automatically launched when a honeyfile is accessed. In fact, when a reader process accesses the trap the system resumes the detection process (writer) and notifies the user to take corrective action. For a quicker and easier response, this task has been semi-automatizated in R-Locker by making use of two lists:1.A *whitelist*, which is created at installation time and contains all legal applications on the system. An application from this list that interacts with the trap is automatically unlocked by the monitor.2.A *blacklist*, which contains programs that the user has already identified as malicious. This list is built as malicious processes are identified, such as when tagged by the user in response to the notification system.

## 4. Lightweight Ransomware Detection in Android Based on inotify()

The Android security model is robust but has some problems that can be exploited by attackers [69]: (i) malicious applications can surpass Google Market controls or are installed from unknown sources; (ii) over-requesting the permissions (seen as vulnerabilities) of many applications breaks the principle of least privilege and the associated functionality is not well understood by users; (iii) native code can be executed outside of the Dalvik machine, thus having less memory protection.

As we have previously discussed, the implementation of R-Locker requires that the operating system supports the creation of FIFOs and some link-type object, either hard or symbolic, in the file system. The first requirement is feasible on Android but, regretfully, the second is not. Android allows the creation of FIFOs in the */data/user/0/...* folder as long as the internal storage is formatted with the type *ext4*. Unfortunately, not all devices have the memory formatted in a file system of this type, regardless of whether they are created through the Java interface or through a native application using JNI [70].

Although this would be quite restrictive due to the sandboxing and the specific mobile device used, it could be enough to extend the R-Locker solution to some models as long as we are able to create symbolic or hard links to the FIFO from the target file system to be monitored. Regretfully, the link-building requirement is more problematic. It should be noted that the file system in which external memory is usually formatted is a variant of the FAT system, known as exFAT (‘Extensible File Allocation Table’) [71], which does not support symbolic links or the creation of FIFO objects. Therefore, to build a solution based on that of the R-Locker proposal is not feasible for this type of memory. While it would potentially be possible to format external memory with a type of file system that allows such mechanisms and is supported on Android, for example *ext4*, not all Android systems allow that. Furthermore, this would break the transparency and simplicity required for R-Locker (R4) when installing said solution, since the user would have to make a backup of their external memory to be able to format it to the new type.

At this point, we need to look for an alternative to the base R-Locker mechanism. Two different approaches appear: continuous dynamic monitoring, which we will call ‘active’, in which information about the operation of the system and applications is collected for subsequent analysis; and another approach in which the monitoring works in a ‘reactive’ way, that is, after instructing the operating system about the events that we want to observe, the monitor waits without consuming resources for the kernel to notify it of the occurrence of said events. Both approaches are analyzed below, but first it is interesting to briefly highlight a common issue: the deployment of the trap files, as follows.

The memory of an Android device is divided into two parts:Internal memory: Memory where private data are stored, that is, those application data (directory */data/*) whose access is under the control of the kernel and the data system (directory */*). This is a part of the memory included in the device.External memory: Public data are stored in this memory, that is, data that can be shared by applications, such as personal data, photographs, documents, etc., and whose access is controlled by the permissions granted to the applications. This memory is made up of part of the memory included in the device (embedded flash memory) and external SD cards.

Regarding the permissions that protect external storage, it is important to indicate that they affect all the files contained. That is, it is not possible to grant permissions for a single application to access a specific directory within this storage but, on the contrary, the grant is for the entire external memory. Allowed permissions (READ_EXTERNAL_STORAGE, WRITE_EXTERNAL_STORAGE, MANAGE_EXTERNAL_STORAGE) give access to (a) the table *MediaStore.Files* that contains an index to all files in media storage, (b) the root directory, and (c) all directories in internal storage except */Android/data* and *sdcardAndroid*.

For all of the above, we propose carrying out the deployment of the trap files in the external memory that can be accessed by different applications installed on the Android device.

### 4.1. Active System Monitoring

As discussed in Section 2, ransomware detection generally involves collecting environment-related parameters that are then processed to determine if the observed behavior is benign or malicious [22,24,27]. For such a gathering process in Android platforms, an application like *AMon* (which stands for ‘Android Monitoring’) is needed [72,73]. Developed by the authors, AMon aims to collect dynamic information on numerous aspects of the operation of the mobile device, such as communications, applications, security status, and interface state. This tool does not require special privileges or *root* access to operate and it collects a larger number of parameters than those usually found in other tools in the literature. In addition, it is feasible to easily add new parameters.

This tool allows multiple applications (e.g., malware detection, access control) supported on the monitoring of dimensions like permissions of installed applications [74], consumption of resources such as CPU usage [75], etc.

Despite the power and flexibility of active monitoring, it presents some limitations:Resources involved: Given the continuous monitoring of a number of features and parameters of the target system, the consumption regime can be high. In the case of AMon, the battery consumption in a normal use regime is 0.4%, which is not very high on its own but is double compared to the consumption of an antivirus application (both consumptions measured in a Samsung S9 phone with a 3000 mAh battery).Offline processing of features: Depending on the detection method used, we can process the data collected inside or outside the phone. In most cases, using ML-based techniques, the processing should be done outside the phone so as not to exhaust the device’s resources, such as the battery.Data consumption: If we need to perform off-device processing, an increase in data traffic and greater use of the communication service are expected, which means greater battery consumption as well.

Due to the above, we finally opted for a reactive ransomware detection solution for Android platforms, where the OS alerts us asynchronously when specific events of interest occur.

### 4.2. Reactive System Monitoring

For our ransomware detection solution in Android, we are going to consider two premises: (a) the necessary interaction of the sample with the file system, (b) noadministrator privileges are required for the tool. According to works like [76], the interface provided by FileObserver() will be considered with this aim.

The FileObserver() class [77] on Android encapsulates the inotify() mechanism of Linux [68]. This mechanism notifies the *kernel* which files/directories are targeted for monitoring by calling inotify_add_watch(), which adds an entry to the watch list for each *instance.inotify*. Each watch list entry contains the path of the file to monitor and a bit mask representing the events to monitor, the function returning an observation descriptor that will be used in subsequent operations such as read().

The advantage of this mechanism over other types of monitoring is that the observation descriptors can be read using an asynchronous I/O mechanism, or controlled by signals, such as select(), poll(), or epoll(), where the kernel tells the watchdog process when an event is ready to be read, freeing the watchdog process from constantly polling the system to see if the event has occurred.

Going back to Android, FileObserver() is an abstract class so that an event handler must be established with onEvent(). Each instance of the class can monitor a path and uses an event mask to specify the changes or actions to report. The list of observable events in Android are shown in Table 1, along with a description of their meaning.

FileOberserver monitors files individually, so if we want to monitor the entire external memory we must recursively traverse the directory tree and establish instances for each file to be monitored. Using the function as stated in the documentation is quite simple, as shown by the (recursive) code snippet in Listing 1. We can see that, after establishing the function to manage the events on the files to be monitored, the calling process will wait for said events to take place.

**Listing 1**. File system monitoring whith FileObserver().public void recursiveFileObserver(File root, List<File> files) {File[] = list = root.listFiles();        if (list != null) {   for (File f : list) {     if (f.isDirectory()) {       FileObserver = new FileObserver(f.getAbsolutePath()) {         @Override         public void onEvent(int event, @Nullable String path);         // Operations to manage the event on the path         FileObserver.startWatching();         files.add(f);         recursiveFilesList(f, files);       }      }    }  }}

#### 4.2.1. FileObserver() Related Patterns Associated to Ransomware

At this point, we propose analyzing the patterns of FileObserver()-related events associated with a typical crypto-ransomware sample in order to model its behavior and, from this, to monitor the processes accessing the file system to conclude, if so, the occurrence of a malicious activity.

The usual operation of ransomware is the encryption of the target files to subsequently delete them to make their unencrypted version inaccessible. This process can be carried out in different ways or stages. Works such as [25,61,78] address this issue and show the file operations involved and the order in which they occur. Three manipulation patterns of both the original and the encrypted files appear:*Overwrite*, where the original file is read and its content overwritten over the encrypted one.*Read–encrypt–delete*, where the original file is first read, then the ransomware creates a new file to store the ciphered content, and finally the file is deleted.*Read–encrypt–overwrite*, which is similar to the previous one but the victim file is deleted and overwritten.

In the next section, we will see how our developed detection tool allows capturing these input/output operation patterns on files using the mechanism provided by FileObserver().

## 5. Experimentation

### 5.1. *FileMonitor* Tool

With the aim of testing the validity of FileObserver() as a reactive mechanism for ransomware detection, a fully functional Android application named *FileMonitor* is developed, whose code is publicly Available online: Github (https://github.com/JA-Gomez-Hernandez/FileMonitor (accessed on 1 April 2024)). This application has a simple user interface, as can be seen in Figure 2: At the bottom are buttons to start/stop monitoring, to check the operation, and to clear the monitoring list; at the top, the dangerous applications installed on the device and a filter of events (as indicated in Table 1) will appear.

The following functional and non-functional requirements are defined for the app:FR1: The user can start/stop monitoring the files.FR2: The user can clear the list of monitored events.FR3: The app will show each type of event recorded and the associated path, it being possible to filter the events by typology.FR4: The app will show the list of dangerous applications with the corresponding permissions.NFR1: Both the interface and the use of the application must be simple and easy to understand.NFR2: The list of events must be displayed in an organized manner. A first level will show a summary of the events on a given file; a second level includes a detailed list of them.NFR3: The list of dangerous applications also has two levels: first, it is displayed in a summary form; second, the details are displayed by clicking on each one.NFR4: File monitoring is possible, even if the application is running in the background.

The operation of the application is simple once it is understood how the FileObserver() class works, although it is necessary to implement it as an activity and a service to be run in the background. The service will perform the monitoring itself, while the activity will be responsible for receiving the data from the service and displaying them to the user. Figure 3 shows the sequence diagram of the actions of both monitoring initialization and event notification.

For the development of the detection system, Java with Android Studio is used as it is the official IDE that offers stability, and it allows developing the graphical interface and the emulation on different devices. The implementation of the service is conventional, as shown in Listing 2, where the use of the START_STICKY indicator must be highlighted so that said service does not stop when it is running in the background.

The service must be declared in the manifest with the corresponding permission (android.permission.FOREGROUND_SERVICE), as it appears in Listing 3. You can also see how the app requests permissions to read/write to the external storage, consults on the existing packages on the platform, and activates the device screen.

**Listing 2**. Service creation by *FileMonitor*.@Overridepublic int onStartCommand(Intent intent, int flags, int startId) {       if (Build.VERSION.SDK_INT >= Build.VERSION_CODES.O) {     createNotificationChannel();   } else {      CHANNEL_ID= "";       Notification notification = new NotificationCompat.Builder(this,    CHANNEL_ID).build();  startForeground(1, notification);       for (File f : observed_files){     Log.i("Service:", f.getAbsolutePath());     observers.add(new singleFileObserver(f));  }  Toast.makeText(this,"Initiating Fileobserver() in the folder" +  root.getAbsolutePath() + "/",Toast.LENGTH_SHORT).show();  return Service.START_STICKY;     // Service "sticky" type to avoid                                       been stopped by the system}

**Listing 3**. Service declaration in background by *FileMonitor*.<\\application>. . .<service android:name="com.filemonitor.test.fileobserver.FileObserverService" android:enabled="true"></service>    </aplication>    <uses-permission ..."android.permission.WRITE_EXTERNAL_STORAGE"/><uses-permission ..."android.permission.WAKE_LOCK"/><uses-permission ..."android.permission.READ_EXTERNAL_STORAGE"/><uses-permission ..."android.permission.FOREGROUND_SERVICE"/><uses-permission ..."android.permission.QUERY_ALL_PACKAGES"/>

The communication between the two components of the application, the service, and the activity is carried out through a LocalBroadManager. Moreover, the Android’s API provides functionality to collect information about the running apps through the PackageManager functions getPackageManager() and getInstalledPackages(). This package returns the installed apps and their permissions, those declared in the manifest as well as those granted. All of this is used to choose the apps with dangerous permissions.

The deployment of the detection tool on a real target scenario is similar to the case of R-Locker for Linux and Windows:First of all, we need to deploy the honeyfiles throughout all the folders in the external memory directory tree. To make them transparent to the user, they will have a name of the form ‘*.<name.extension>*’, where ‘.’ is the conventional way to hide files. Both the name and the extension of the file should be attractive to ransomware.When access to some of the monitored files is detected by FileObserver(), we must deploy the necessary countermeasures intended to allow, on the one hand, to determine which application presents the observed malicious behavior and, on the other, to stop said process by notifying the user as we already did in R-Locker.     For that, a whitelist is used, where the applications installed at the time *FileMonitor* application is installed are stored in the whitelist as benign applications. This is done by using the QUERY_ALL_PACKAGES permission to collect the list of applications installed on the device before starting the monitoring process. When the ransomware detection occurs, we collect this information again and locate the app (or apps) installed after the monitoring app and that is (or are) the candidate for a malicious behavior. Following the R-Locker scheme, we can kill said process by giving our application the KILL_BACKGROUND_PROCESSES permission. Unlike the detection proposal for Linux and Windows, the one for Android is capable of building a more selective whitelist, since we can select only applications that have the necessary permissions to cause damage to external memory.    In addition to the whitelist, we can use, as we already did in R-Locker, the blacklist solution with the objective of detecting and (semi-)automatically stopping different occurring threads of an already known ransomware sample.

### 5.2. Test Scenario and Results

To carry out experimentation with our Android ransomware detection proposal through the *FileMonitor* app, the following execution environment is deployed:A virtual machine with VirtualBox, as a sandbox.An Android virtual machine inside the previous VM. This second level of virtualization is made with GenyMotion [79].

In addition, the malware samples shown in Table 2, which were obtained from the AndroidMalware repository [80], are also considered for experimentation. It is important to mention that there are not only ransomware samples but also other types of malware.

Figure 4 depicts the operation captured for the Cyberpunk ransomware sample (an app posing as a fashion game of the same name, allowing the user to grant it write permissions), where the behavior ‘read–encrypt–delete’ previously mentioned in Section 4.2.1 is observed. A similar situation occurs for the sample named Crydroid. As the mentioned figure shows, the files undergo three operations:1.A DELETE event, which indicates a file erase operation. Here, file erase (unlink()) means that the information on disk is not really deleted: the file directory entry is deleted but data remains intact since there is an active reference to the file in the kernel (while the file is open). This way, ransomware can access the file but the user cannot. In Figure 4, this is the first operation (see timestamp) on the file ‘Picture98.png’.2.Second, the creation of the encrypted version of the file, which usually has the same name and a specific extension (CREATE event). In our case, this operation will generate the encrypted version of the target file: ‘Picture98.png.CoderCrypt’.3.A MODIFY event, which reflects that the content of the file is modified, including a potential change in its extension. Finally, the ciphered version of file is written.

The abovementioned behavior fits into the type of pattern described in Section 4.2.1, due to the order in which the ransomware performs actions on the victim file. Therefore, the detection application would allow detection of ransomware samples running on devices by characterizing the behavior of said processes during the encryption procedure.

Beyond the correct detection of the analyzed ransomware samples, some measurements were taken to determine the response time of the notification mechanism implemented in *FileMonitor*: monitoring 200 events for a target file requires 1–200 ms if only one application is running, 1–450 ms if four apps are executed, and 1–700 when nine apps are running. Moreover, the cost of encrypting a file has also been measured: 10 ms for an 8 KB file; 70 ms for a 60 KB file. Therefore, a ransomware sample could encrypt about ten files if we distribute 60 KB decoys, so we can consider that it behaves as an early detection mechanism.

#### Additional Malware Detection Capabilities

An advantage of our tool *FileMonitor* is that it can detect other malicious behaviors affecting filesystems besides those specific to crypto-ransomware. For example, the spyware ThiefBot, which appeared in September 2020, is intended to obtain banking credentials, although it collects all types of credentials and personal information. This app impersonates the Google Play app and requests permissions to access storage, SMS, phone, contacts, and camera. When installing the app, it displays an error message, indicating that it is not able to work on older versions of Android, and that newer versions of the app are needed. Then it seems to close, when it really starts working (spying) in the background.

If we use *FileMonitor* after installing this spyware, we obtain the results shown in Figure 5, where we can see a very high number of accesses of type ACCESS and OPEN to a folder created by the malicious app, named *downloads*. As shown, the folder is accessed in a short period of time with a very high number of events (in the order of 5000 events per minute). This makes us suspicious of its malicious behavior.

### 5.3. Discussion

After testing our detection solution for Android platforms, we can conclude that it behaves similarly to R-Locker for Linux/Windows platforms from the perspective of the desired features, F, and requirements, R, as well as the figures of accuracy and efficiency achieved. In addition, it presents capacities to detect other malware typologies that rely on file access.

It is important to mention that only a few similar works in the literature show specific figures on the efficiency of the solution (e.g., see [35,41,44] in Section 2). Despite the good performance achieved by ours, further investigation needs to be expanded with additional ransomware/malware samples. Regretfully, it is a difficult task that would require more time and dataset resources.

A relevant issue to be further investigated is the one regarding the specific file access sequences to conclude actual harmful file accesses. Otherwise, false alerts will be generated. Fortunately, the number of access patterns to manipulate files using the functions offered by the operating system is limited and, therefore, well characterized.

## 6. Conclusions

This work discusses Android ransomware detection and introduces a new lightweight detection approach for that platform. Unlike most current detection proposals, which rely on usually complex and heavy ML-based detection schemes, our solution relies on a reactive file system monitoring methodology based on the deployment of honeyfiles, which is expected to involve low resource consumption in terms of memory, CPU, and battery.

Implemented through a specific app named *FileMonitor* to experimentally evaluate the efficacy and efficiency of the proposed methodology, the results obtained show the progress that our solution provides is state-of-the-art, making it a good candidate to be adopted to effectively combat ransomware on small and resource-limited platforms.

As shown, our proposal is not only capable of detecting ransomware samples but is also able to detect potential harmful actions against files by simply analyzing and detecting the sequence of certain operations on specified targets files. This way, the solution will depend on the good characterization of the sequences and frequency of the operations carried out by ransomware.

Despite the quality of the proposal and its promising use in mobile environments (e.g., BYOD security policies), it should be experimentally evaluated in a more exhaustive way by using more crypto-ransomware and other malware-targeting file samples. Moreover, in the present form, the developed app has the file access patterns considered for detection embedded in the code. Therefore, a quick improvement would consist of allowing the inclusion of new patterns through a configuration file or a similar scheme.

Finally, the solution introduced is designed to be deployed as an autonomous, separate tool. However, its integration with other security frameworks and solutions should be analyzed.

## Figures and Tables

**Figure 1 sensors-24-02679-f001:**
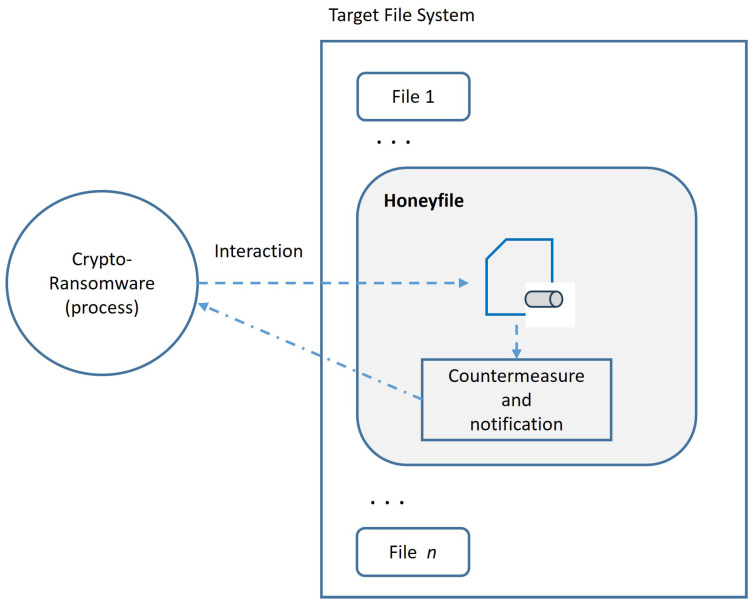
General architecture of the R-Locker proposal.

**Figure 2 sensors-24-02679-f002:**
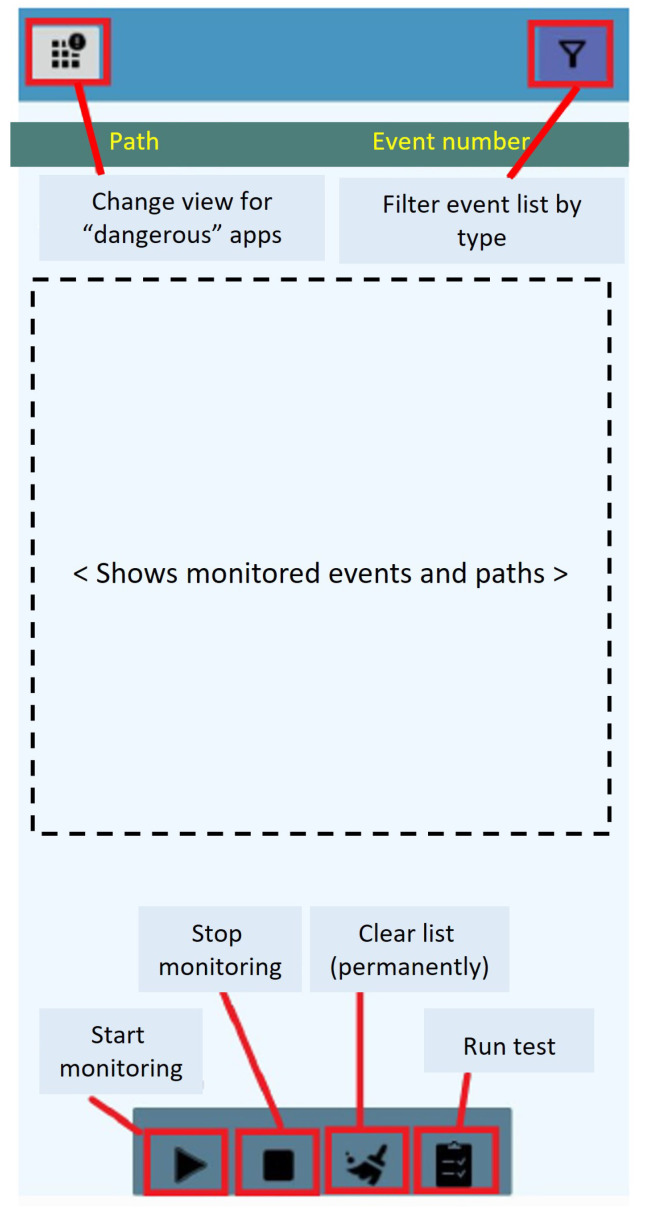
Graphical interface of the *FileMonitor* app.

**Figure 3 sensors-24-02679-f003:**
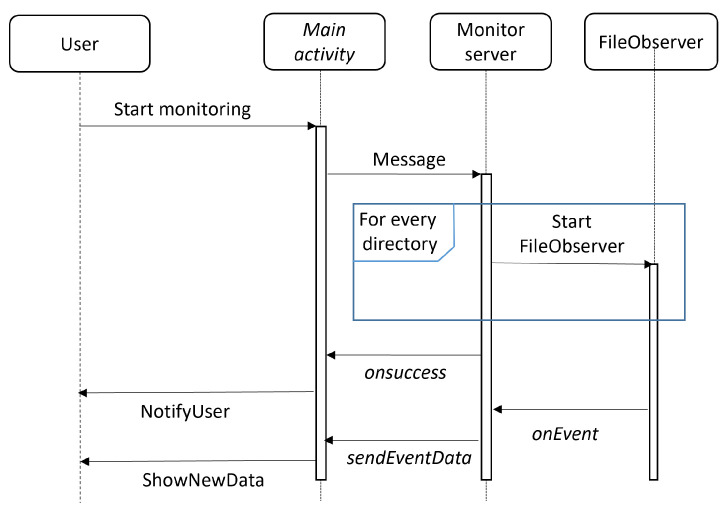
Sequence diagram of the monitoring initialization and event notification.

**Figure 4 sensors-24-02679-f004:**
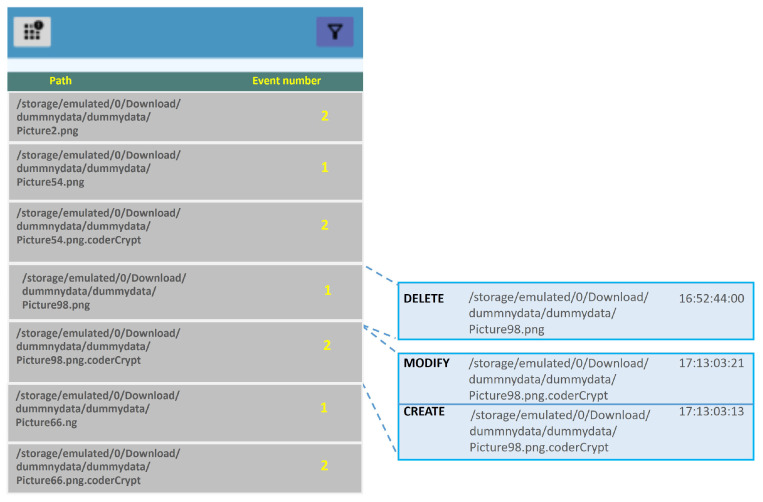
Monitoring the file encryption process by the CyberPunk ransomware sample with *FileMonitor*.

**Figure 5 sensors-24-02679-f005:**
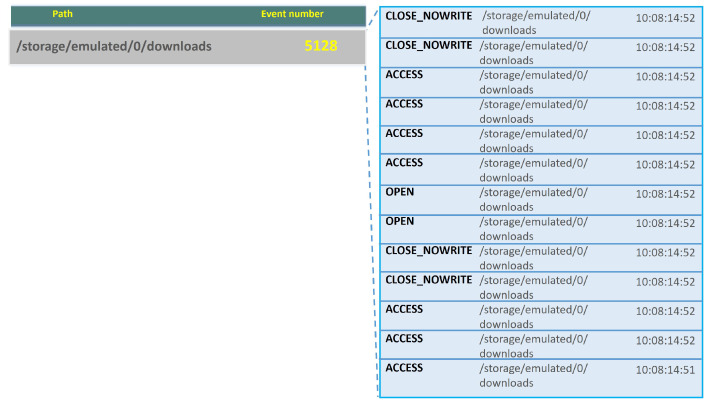
Monitoring results obtained by *FileMonitor* for the sample ThiefBot.

**Table 1 sensors-24-02679-t001:** Monitorable events with FileObserver().

Event	Description
ACCESS	Data has been read from the file
ALL_EVENTS	All valid event types combined
ATTRIB	They have changed the metadata explicitly
CLOSE_NOWRITE	Someone has file/directory open read-only and closed it
CLOSE_WRITE	Someone has file/directory open for writing and closed it
CREATE	A file/directory has been created in the monitored folder
DELETE	A file has been deleted from the monitored directory
DELETE_SELF	The file/directory in the monitored directory has been deleted; stop monitoring
MODIFY	File data has been written
MOVED_FROM	A file or directory was moved from the monitored folder
MOVED_TO	A file or directory was moved to the monitored folder
MOVED_SELF	The observed file or directory has been moved; monitoring continues
OPEN	A file or directory has been opened

**Table 2 sensors-24-02679-t002:** Malware samples used for experimentation, from [80].

Name	Hash MD5	Type
CookierStealer	65a92baefd41eb8c1a9df6c266992730	Spyware
Covid_SpyPhone	3288a6cb81bc3e928e438fa280fec847	Riskware
Covid_Cerberus	66c4513025128719dda018820cc0987e	Spy/Dropper
Crydroid	381134ea0f0be535b9d2ce8a94093576	Ransomware
Cyberpunk	cbd92757051490316de527a02ac17947	Ransomware
Joker	44faa3de0f17491557a3a775c88e7e33	Spy/Dropper
Shopaholic	0a421b0857cfe4d0066246cb87d8768c	Dropper
ThiefBot	e88867956017bbe5b633811885c87018	Spyware
Trickbot	05c0c1bb23cc06474c3fd3ba51e4e4c6	Spy/Dropper

## Data Availability

No datasets were generated during the current study. The cited datasets are described in the corresponding publication of the bibliography.

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
