# Peer review of "Lightweight Crypto-Ransomware Detection in Android Based on Reactive Honeyfile Monitoring"

_sensors, 2024, doi:10.3390/s24092679_

Round 1

Reviewer 1 Report

Comments and Suggestions for Authors

Section – wise report:

1.      Abstract

The abstract mentions the utilization of honeyfiles combined with a reactive monitoring scheme. It could further clarify how this approach is novel compared to existing methods. A brief mention of the unique aspects of your solution, such as specific adaptations for Android's constraints, would be beneficial.

2.      Introduction:

The introduction could benefit from a brief overview of the proposed solution's unique contributions to the reader's anticipation. More statistics or recent examples of ransomware attacks could enhance the argument's impact.

3.      Background of Ransomware Detection for Android Platforms:

The discussion on static vs. dynamic detection is informative but could be enhanced by directly comparing these methods' effectiveness and resource consumption to highlight the gap your research fills. Some subsections could be condensed to maintain focus on aspects most relevant to the proposed solution.

4.      Ransomware Detection Based on Honeyfiles:

This section is well-developed but could benefit from clearer distinctions between general honeyfile approaches and the unique aspects of R-Locker. Consider emphasizing more on the limitations of existing honeyfile approaches that your Android solution aims to overcome.

5.      Lightweight Ransomware Detection in Android Based on inotify():

More details on the implementation challenges specific to Android and how they were overcome would be beneficial. A comparison with R-Locker’s performance on Linux/Windows could help highlight the adaptation’s significance.

6.      Experimentation:

While the results are promising, a more detailed analysis comparing these results with existing solutions' performance could offer a clearer picture of the proposed solution's advantages. Consider addressing potential scalability issues or limitations encountered during testing.

7.      Conclusion:

Any research has limitations, and briefly acknowledging these in the conclusion can provide a balanced view of your work. This section could be enhanced by briefly mentioning any immediate practical applications or ongoing collaborations to integrate this solution into existing security frameworks. A reflection on the broader implications of this research for mobile security would be a meaningful addition.

Author Response

  • Comment 1:

 The abstract mentions the utilization of honeyfiles combined with a reactive monitoring scheme. It could further clarify how this approach is novel compared to existing methods. A brief mention of the unique aspects of your solution, such as specific adaptations for Android's constraints, would be beneficial.

 Action/answer:

According to the reviewer’s suggestion, main aspects of the detection solution are now mentioned in the Abstract.

  • Comment 2:

 The introduction could benefit from a brief overview of the proposed solution's unique contributions to the reader's anticipation. More statistics or recent examples of ransomware attacks could enhance the argument's impact.

 Action/answer:

Again, some additional relevant aspects of the solution are included in Section 1-Introduction (see page 2).

  • Comment 3:

The discussion on static vs. dynamic detection is informative but could be enhanced by directly comparing these methods' effectiveness and resource consumption to highlight the gap your research fills. Some subsections could be condensed to maintain focus on aspects most relevant to the proposed solution.

 Action/answer:

Static and dynamic detection schemes present particular strengths, so that they can be used separately or in combination, as discussed  throughout Section 2-Background. In any case, both constitute active detection schemes whose resource consumption is greater than that involved in our reactive detection approach. This is briefly highlighted at the end of the last paragraph in Section 2-Background.

  • Comment 4:

This section is well-developed but could benefit from clearer distinctions between general honeyfile approaches and the unique aspects of R-Locker. Consider emphasizing more on the limitations of existing honeyfile approaches that your Android solution aims to overcome.

 Action/answer:

The principal benefit of R-Locker compared to other solutions is stated in Section 3.3.1 through features F1 and F2. We appreciate the comment and thus we emphasize this aspect in the text (see the end of first paragraph in Section 3.1.1, page 6).

  • Comment 5:

More details on the implementation challenges specific to Android and how they were overcome would be beneficial. A comparison with R-Locker’s performance on Linux/Windows could help highlight the adaptation’s significance.

 Action/answer:

We thank the reviewer for the comment, but we think that this aspect is properly dealt with in the current version of the paper.

After presenting the R-Locker basics throughout Section 3.1, we discuss in Section 4 (see second and third paragraphs) the main challenges for implementing R-Locker on Android platforms (i.e., FIFOs and link objects). Then, after introducing the reactive methodology considered for Android in Section 4.2, Section 5 shows and analyzes the detection results achieved, which are similar to those obtained in Linux/Windows platforms from the perspective of the desired features F and requirements R, as well as the accuracy and efficiency achieved.

All of that is briefly discussed in a new Section 5.3, with the aim of highlighting the main contributions and limitations of our solution.

  • Comment 6:

While the results are promising, a more detailed analysis comparing these results with existing solutions' performance could offer a clearer picture of the proposed solution's advantages. Consider addressing potential scalability issues or limitations encountered during testing.

 Action/answer:

This is also pointed out in the new Section 5.3.

  • Comment 7:

Any research has limitations, and briefly acknowledging these in the conclusion can provide a balanced view of your work. This section could be enhanced by briefly mentioning any immediate practical applications or ongoing collaborations to integrate this solution into existing security frameworks. A reflection on the broader implications of this research for mobile security would be a meaningful addition.

 Action/answer:

We agree with the reviewer… See new Section 5.3 and some additional comments in Section 6.

  • Additional reviewer’s comments…

1) The authors should consider extending their experimental evaluation to include a broader range of ransomware and other malware samples. This could provide a more comprehensive assessment of the detection tool's effectiveness across different attack vectors.

Action/answer:

Again, see new Section 5.3.

2) Incorporating a comparative analysis with existing detection methods would offer a clearer picture of the proposed solution's performance in terms of detection accuracy, speed, and resource efficiency. Providing deeper technical insights into the use of inotify() and its implications for monitoring efficiency and potential impacts on device performance would enhance the reader's understanding of the solution's practicality.

Action/answer:

See new Section 5.3.

Reviewer 2 Report

Comments and Suggestions for Authors

You have effectively identified a crucial gap in ransomware mitigation strategies for mobile devices, proposing a novel solution that is both resource-efficient and practical for widespread adoption. 

This strategy reduces the resource demands typically associated with machine learning-based ransomware detection methods.

-The effectiveness of the proposed solution against advanced ransomware variants that might bypass honeyfile detection remains uncertain. The paper could benefit from a broader discussion on integrating this solution within existing cybersecurity frameworks to enhance overall system security.

- The revisions are recommended to address the concerns about the solution's robustness against advanced ransomware techniques and its testing in varied operational environments.

-Broadening the scope of your testing to include a diverse range of ransomware samples and operational settings will strengthen your findings. A more varied dataset could help simulate a wider array of attack vectors, offering a deeper insight into the tool's effectiveness across different scenarios.

-An in-depth discussion on potential evasion tactics by ransomware and how your tool can adapt to these challenges would provide a more balanced view of its capabilities and limitations.

-Exploring how your tool can be integrated with existing security measures and systems would highlight its practicality and potential for real-world application. This discussion could also include user experience considerations, particularly in relation to false positives and system disruptions.

Comments on the Quality of English Language

none

Author Response

  • Comment 1:

 The effectiveness of the proposed solution against advanced ransomware variants that might bypass honeyfile detection remains uncertain. The paper could benefit from a broader discussion on integrating this solution within existing cybersecurity frameworks to enhance overall system security.

 Action/answer:

See new Section 5.3.

  • Comment 2:

 The revisions are recommended to address the concerns about the solution's robustness against advanced ransomware techniques and its testing in varied operational environments.

 Action/answer:

Again, see new Section 5.3 and some additional comments in Section 6.

  • Comment 3:

 Broadening the scope of your testing to include a diverse range of ransomware samples and operational settings will strengthen your findings. A more varied dataset could help simulate a wider array of attack vectors, offering a deeper insight into the tool's effectiveness across different scenarios.

 Action/answer:

Once more, see new Section 5.3 and some additional comments in Section 6.

  • Comment 4:

 An in-depth discussion on potential evasion tactics by ransomware and how your tool can adapt to these challenges would provide a more balanced view of its capabilities and limitations.

 Action/answer:

As mentioned in Sections 5.3 and 6, a key point of our solution is to define the specific access sequences on the traps to be detected. Fortunately, it should be noted that this number of sequences is  limited. This way, alternative evasion methods (like RIPlace [1,2], below) use file related operations (open, read, write) that can be detected by our method.

In the present work, the reference [1] below is included as [83] to talk about potential evasion techniques in Section 4.2.1.

[1] K. Begovic, A. Al-Ali, Q. Malluhi: “Cryptographic ransomware encryption detection: Survey”, Computers & Security, Volume 132,2023, 103349 (DOI: 10.1016/j.cose.2023.103349).

[2] CISOMAG. RIplace – A Security Evading Ransomware Technique. CISO

MAG, Cyber Security Magazine 2019. https://cisomag.eccouncil.org/riplace- ransomware-technique/ (accessed January 10, 2022).

  • Comment 5:

Exploring how your tool can be integrated with existing security measures and systems would highlight its practicality and potential for real-world application. This discussion could also include user experience considerations, particularly in relation to false positives and system disruptions.

 Action/answer:

See new Section 5.3 and some additional comments in Section 6.

Reviewer 3 Report

Comments and Suggestions for Authors

Based on previous authors’ work [32,33], this paper presents a new ransomware detection tool for Android platforms with three main contributions. By installing some honeyfiles and monitoring accesses of these files, it can detect ransomeware reactively. The strong point is it is  lightweight and does not require modification of kernel. They implemented using FileObserver(), which uses inotify() to monitor the access of the specific files, honey files. Another strong point is they opened their source code in GitHub.

English writing is understandable, but somewhat revised/compacted to clearly show relevant things. Especially from page1-10. The proposed scheme description is only one-page, Section 4.2. In related work section, the tense of the sentences must be past.

Author Response

  • Comment 1:

English writing is understandable, but somewhat revised/compacted to clearly show relevant things. Especially from page 1-10. 

Action/answer:

The document has been reviewed to remark relevant aspects.

  • Comment 2:

The proposed scheme description is only one-page, Section 4.2.

Action/answer:

Although the particularities of the detection scheme for Android are mainly discussed in Section 4.2, the overall functionally and design provided in Section 3.1 are also part of to the solution.

  • Comment 3:

In related work section, the tense of the sentences must be past.

Action/answer:

Based on the reviewer’s suggestion, this aspect has now been fixed in the new version of the manuscript!

Round 2

Reviewer 2 Report

Comments and Suggestions for Authors

None

Comments on the Quality of English Language

none